# First Report of *Diaporthe goulteri* on Soybean in Germany

**DOI:** 10.3390/jof10110803

**Published:** 2024-11-20

**Authors:** Behnoush Hosseini, Maximilian Gerhard Gröbner, Tobias Immanuel Link

**Affiliations:** Department of Phytopathology, Institute of Phytomedicine, Faculty of Agricultural Sciences, University of Hohenheim, 70599 Stuttgart, Germany; behnoush.hosseini@uni-hohenheim.de (B.H.); maximilian.groebner@uni-hohenheim.de (M.G.G.)

**Keywords:** isolation, species identification, phylogeny, morphology, pathogenicity

## Abstract

*Diaporthe* (anamorph: *Phomopsis*) species are endophytes or fungal pathogens for many different plant species. Soybean (*Glycine max*) can be infected by many different *Diaporthe* species; among them, *D. caulivora* and *D. longicolla* are responsible for the most significant damages. *Diaporthe goulteri* is a species that was only recently described and has so far been found on sunflower (*Helianthus annuus*) in Australia and an unknown host in Thailand. Here, we report isolation of *D. goulteri* from soybean in southern Germany, molecular species identification, and additional morphological description. We also show that *D. goulteri* can infect soybean and describe the symptoms we observed, both on the plant where the isolate came from and following artificial inoculation.

## 1. Introduction

Because of increased demand for soybean (*Glycine max*) as a source for oil and protein, its cultivation is increasing in Germany like in other countries. As soybean cultivation area is on the rise, it is expected that the incidence of soybean pathogens will increase. Species of the fungal genus *Diaporthe* are important pathogens for soybean in all major soybean cultivation areas in North and South America, Asia, and Europe [1,2,3,4,5,6]. *Diaporthe longicolla* was identified as a seed decay pathogen on German soybean seeds together with *D. caulivora*, *D. eres*, and *D. novem* [7]. Since globally many more *Diaporthe* spp. are reported as soybean pathogens [8], the occurrence of more *Diaporthe* spp. on soybean in Germany was expected.

Genus *Diaporthe* is large and complex, containing more than one hundred species. The phylogeny of the genus has been revised based on sequence comparisons [9,10,11] and studies concerned with updating the molecular phylogeny of *Diaporthe* regularly discover new species belonging to the genus. *Diaporthe goulteri* is among those species discovered relatively recently: Thompson et al. [12] isolated the species from a sunflower (*Helianthus annuus*) seed, described it, and named it. Apart from this, there are scant reports about the species; there are some isolates from China, Taiwan, and Thailand with corresponding sequences deposited in GenBank, but we found only one publication [13] that provides additional information about the species: Bundhun et al. [13] reported the isolation of *D. goulteri* from “a dead branch of an unknown host” in Thailand and described its sexual morph.

Here, we report isolation of *D. goulteri* from soybean in southern Germany. We provide evidence that soybean is a host plant for *D. goulteri* and expand on the species description.

## 2. Materials and Methods

### 2.1. Sampling and Isolation

Soybean plants showing signs of *Diaporthe* infection, i.e., pycnidia or perithecia showing up as black dots on the stem were collected from fields shortly before harvest. Seeds, pods, and stem pieces were disinfected with 3% sodium hypochlorite solution for 1 min, rinsed three times with sterile water, dried on filter paper, and placed on Petri plates with potato dextrose agar (PDA). These plates were incubated at room temperature for one week. After seven days, cultures, which were tentatively identified as *Diaporthe* sp. based on culture morphology, were subcultured by transferring an agar plug with a small piece of the cultures to fresh PDA plates that were also incubated at room temperature.

### 2.2. Species Identification

#### 2.2.1. Species Identification from Pure Cultures Using Culture Morphology and Microscopy as Well as Sequencing and Phylogenetic Analysis

Approximately 20 or 40 days-old cultures were evaluated morphologically. This way, it was decided for which cultures molecular species identification should be carried out. The data also now contribute to the species description. The cultures on PDA were observed from the top and from the bottom (front and back side of the plates). Conidiomata on the plates and pycnidia on soybean stems were observed using a Stemi 2000 binocular loupe (Carl Zeiss, Oberkochen, Germany) and conidia using a Primo Star microscope (Carl Zeiss). The images were taken using an AxioCam HRC color camera (Carl Zeiss) and evaluated with AxioVision software, Release 4.8.3 Special Edition 1 (Carl Zeiss, Oberkochen, Germany).

For molecular species identification, genomic DNA was extracted from 7 days-old cultures using the protocol published by [14]. ITS, *TEF1*, and *TUB* genes were amplified using the primer pairs ITS1-F/ITS4 [15], EF1-728F/EF1-986R [16], and Bt-2a/Bt-2b [17] under the same conditions as described by [7] and sequenced. The sequences were then searched against NCBI GenBank using BLASTn (https://blast.ncbi.nlm.nih.gov/Blast.cgi?PROGRAM=blastn&PAGE_TYPE=BlastSearch&BLAST_SPEC=&LINK_LOC=blasttab&LAST_PAGE=blastn accessed on 10 October 2024).

For the isolate (DPC_HOH36) with high sequence similarity to *D. goulteri*, we downloaded highly similar sequences from GenBank and performed phylogenetic analysis. For downloading, we chose >90 percent identity as a criterion and cutoff. The sequences were aligned using ClustalW [18] and the alignments were edited using BioEdit version 7.0.5.3 [19]. For the three-gene phylogeny, the alignments were concatenated using a text editor. Maximum likelihood phylogenetic analysis using the Tamura–Nei model [20] was performed using MEGA X version 10.0.5 [21]. Initial trees were obtained automatically by applying the maximum parsimony method.

#### 2.2.2. Direct Species Identification from Infected Plant Parts Using qPCR

For qPCR detection, pieces of leaves, stems, pods, or seeds were surface disinfected as described in Section 2.1 and used for DNA isolation. Samples of roughly 100 mg were put into 2-mL micro screw tubes (Sarstedt, Nümbrecht, Germany) with two steel balls (4.50 mm in diameter, Niro, Sturm Präzision GmbH, Oberndorf am Neckar, Germany). The tubes with the samples were frozen in liquid nitrogen for 3 min and homogenized for 20 s at 4.5 m/s using a FastPrep^®^-24 homogenizer (MP Biomedicals GmbH, Santa Ana, CA, USA). DNA extraction was performed using the DNeasy Plant Mini Kit (Qiagen, Hilden, Germany) following the manufacturer’s instructions.

Field samples were tested for the presence of *D. caulivora*, *D. eres*, *D. longicolla*, or *D. novem*. The reactions for the quadruplex TaqMan qPCR to detect these species were prepared as previously described [22].

The 20-μL reactions used here for detection of *D. goulteri* in plant samples consisted of 10 μL ProbeMasterMix (2×) No-ROX (Genaxxon bioscience GmbH, Ulm, Germany), 4 pmol of each forward and reverse primers (DPCG-F: 5′-cttacactcacaaaactcgc-3′; DPCG-R: 5′-gctcgattcaccgggttg-3′), 1 pmol probe (DPCG-P: 5′-6-FAM-ccagagcaaacaccaccgacgc-BMN-Q535-3′), and 2 μL template DNA. Design and testing of this primer and probe combination was performed analogous to what was described previously [22]. The reaction mixes were incubated for 3 min at 95 °C and then subjected to 40 cycles of 95 °C for 15 s and 60 °C for 45 s. Reactions were run in technical duplicates on a CFX96 Real-Time PCR system (Bio-Rad Laboratories, Hercules, CA, USA) using FrameStar1 96-Well Skirted PCR Plates (4titude, Brooks Automation, Chelmsford, MA, USA).

### 2.3. Testing for Pathogenicity on Soybean

Inoculations for the pathogenicity test were performed using the toothpick method [23,24]. To prepare toothpicks for inoculation, autoclaved toothpicks were placed on PDA plates together with agar plugs overgrown with mycelium of the *Diaporthe* isolate. The plates were incubated at room temperature for 20 days. Toothpicks overgrown with mycelium were inserted into the stem in the middle of the internode between cotyledons and the first trifoliate of three weeks old healthy soybean plants (cv. Shouna) at an angle of 90 degrees. Control plants were identically treated with sterile toothpicks. Six plants each were used for inoculation and control plants were kept in the greenhouse at 16 h light / 8 h dark (24 °C/22 °C); air humidity was kept high using a vaporizer (Condair, Norderstedt, Germany). Symptom development was observed weekly for up to two months.

The inoculated plants were tested for infection with *D. goulteri* by two different methods. One was re-isolation, and the other was detection using qPCR with a newly designed primer–probe combination for species-specific detection based on the *TEF* sequence of *D. goulteri*, as described in Section 2.2.2.

For re-isolation, the stem of an infected plant was cut close (1 cm removed) to the inoculation site. After surface disinfection (as described in Section 2.1), 2 to 3 mm long and 1 mm thick pieces of the center part of these sections of the stems were placed on PDA plates. The developing fungal colonies were determined as identical to the inoculum based on culture morphology to fulfil Koch’s postulate.

For qPCR detection, pieces of the center part of the above sections of the stems of the infected plant close to the inoculation site (treated as for re-isolation) and identical stem pieces from a control plant were used for DNA extraction. DNA extraction and qPCR were performed as described in Section 2.2.2.

## 3. Results

### 3.1. D. goulteri Found on a Soybean Plant Collected in a Field Close to Tübingen, Germany

As part of a survey monitoring the incidence of *Diaporthe* spp. in Germany, soybean plants showing typical *Diaporthe* symptoms were collected from fields close to Tübingen on 2 October 2023. Using the qPCR assay developed by [22], plants infected with *D. caulivora*, *D. eres*, *D. longicolla*, or *D. novem* were identified (78% of all sampled plants); the highest incidence was found for *D. caulivora*. From some plants, especially those that tested negative in the qPCR assay, parts were also used for gaining *Diaporthe* spp. isolates (see Section 2.1). This way, we gained five new isolates of *Diaporthe* spp.; the other samples either yielded nothing or fungi from different genera. One isolate (designated DPC_HOH36) recovered from a seed of a plant collected from the following coordinates (48.525801, 9.096385) showed high similarity in all three tested genes to sequences annotated with *D. goulteri* in the GenBank database. This, together with morphological observations, allowed for the identification of isolate DPC_HOH36 as *D. goulteri*.

To refine this finding, similar sequences were downloaded from GenBank. Because of the chosen cutoff, the download was restricted to sequences annotated with either *D. goulteri* or *D. ambigua*. To broaden the phylogenetic analysis we also included sequences representing the four *Diaporthe* spp. reported previously on soybean in Germany (*D. caulivora*, *D. eres*, *D. longicolla*, and *D. novem*) [7] as outgroups (Figure 1).

For all three genes, and also in the analysis combining the three genes, our sequences clustered with the sequences annotated as *D. goulteri* in GenBank. The ITS sequence of DPC_HOH36 is identical to that of strain BRIP 55657a, the isolate corresponding to the first description of *D. goulteri* [12] (Figure 1b). The same is true for the *TUB* sequence (Figure 1c), while the *TEF* sequence of DPC_HOH36 is identical to that of isolate MFLUCC 21-0012 [13] (Figure 1d). Additional sequences annotated with *D. goulteri* were found and are represented in the phylogenies; however, no publications were found related to the GenBank records. Altogether, the analysis of our sequence data clearly allowed for the identification of isolate DPC_HOH36 as *D. goulteri*.

Isolates and corresponding GenBank accession numbers used for the phylogenies are detailed in Table 1.

### 3.2. Morphological Description of D. goulteri Isolate DPC_HOH36

Growth of *D. goulteri* isolate DPC_HOH36 on PDA was fast compared to other *Diaporthe* spp. (*D. longicolla*, *D. caulivora*, *D. eres*, and *D. novem*) and after 5 days the entire plate was covered with mycelium of white-yellowish-light green color. The back side of the plates was white and after longer incubation (40 days) it became light brown with scattered dark spots (Figure 2b).

After approximately 40 to 50 days, dark brown pycnidia appeared (Figure 2a,c) that contained numerous α-conidia (Figure 2e). The α-conidia were fusiform to cylinder-shaped, biguttulate, hyaline, and measured 5.19 to 8.06 × 1.92 to 2.88 µm. Perithecia and β-conidia were not observed. The morphological characteristics of isolate DPC_HOH36 were similar to those described for *D. goulteri* strain BRIP 55657a [12]. Thus, the morphology corroborated the identification of isolate DPC_HOH36 as *D. goulteri*.

### 3.3. D. goulteri Can Infect Soybean in the Laboratory Test

We used an established method for isolating fungi growing inside plant tissue to produce our isolates (see Section 2.1). Therefore, our isolate should be either an endophyte or a pathogen. To further establish whether *D. goulteri* can infect soybean, which parts of the plant are affected, and to observe symptoms caused by *D. goulteri*, we performed a pathogenicity test, as described in Section 2.3.

Dark discoloration of the plant tissue extending roughly 1 cm from where the inoculated toothpicks had been inserted was observed at the soybean stem (Figure 3c). When the stems were cracked open, this discoloration, indicating necrosis, could also be found in the center of the stem, where it extended to roughly 2 cm from the inoculation site (Figure 3d). In contrast, control plants that had been stuck with sterile toothpicks showed a tan discoloration that extended less than 2 mm from the insertion site.

Based on culture morphology, fungal mycelium originating from re-isolation from the discolored plant parts was identified as *D. goulteri*. Using qPCR, we also could detect *D. goulteri* in the stem of inoculated plant close to the infection site (Figure 3e). No *D. goulteri* was detected on any sample taken from control plants (Figure 3f). These observations strongly indicate that *D. goulteri* can infect soybean plants.

## 4. Discussion

Strain DPC_HOH36 was isolated from a soybean plant in a field close to Tübingen, Germany. BLAST comparisons of the *TUB*, *TEF*, and ITS sequences showed high similarity to published sequences of *D. goulteri*. This similarity can also be seen in phylogenies built based on these sequences. These results strongly suggest that our isolate DPC-HOH36 is *D. goulteri*.

Thompson et al. [12] described *D. goulteri* colonies growing on PDA and OMA (oatmeal agar) with relatively fast-growing mycelium that was first white before dark spots appeared. On PDA, we made the same observation. Droplets containing conidia formed on OMA were quite colorful—pale yellow, orange, or ochre and reddish, or as the authors described it, “sienna colored”. It is probably due to the medium that we could not observe these colors on our PDA plates. The form of the conidia they observed was exactly as seen by us, and with 6 to 9 × 2 to 3 µm, the dimensions they observed also encompass what we measured. The conidiomata observed by [12] on OMA and on sterilized wheat straw are slightly different from what we observed on PDA and on sterilized soybean stems. Nevertheless, morphological observations also indicate that isolate DPC_HOH36 is the same species. Our description of the species should be seen as an extension of the existing descriptions.

Thompson et al. [12] isolated *D. goulteri* from a sunflower seed. No further tests regarding the pathogenicity of *D. goulteri* on sunflower or on association to any other plant species were performed. The species from which other researchers isolated *D. goulteri* are either unidentified or not mentioned [13]. Therefore, the host range of *D. goulteri* remains largely unknown. Our pathogenicity test on soybean is the first demonstration of infection of a plant by *D. goulteri*, so it can be stated that *G. max* definitely is a host for *D. goulteri.*

We performed surface disinfection of the plant material before isolation of *Diaporthe* spp. Using this procedure, no growth of fungi only randomly associated (i.e., spores sticking to the surface) with the sampled soybean plants should occur. Therefore, isolating *D. goulteri* from a symptomatic soybean plant means that the fungus was growing inside the plant, which fulfils the first of Koch’s postulates and indicates that it is either an endophyte or a pathogen. Together with that, inoculation, infection and detection of the fungus in the inoculated soybean tissue demonstrated that *D. goulteri* can infect and grow in soybean. This constitutes the first demonstration of the ability of *D. goulteri* to infect plants. More findings of *D. goulteri* in the field together with observations of symptoms and damages will be necessary to establish whether *D. goulteri* acts as a pathogen and how severe damages caused by the species may be.

## Figures and Tables

**Figure 1 jof-10-00803-f001:**
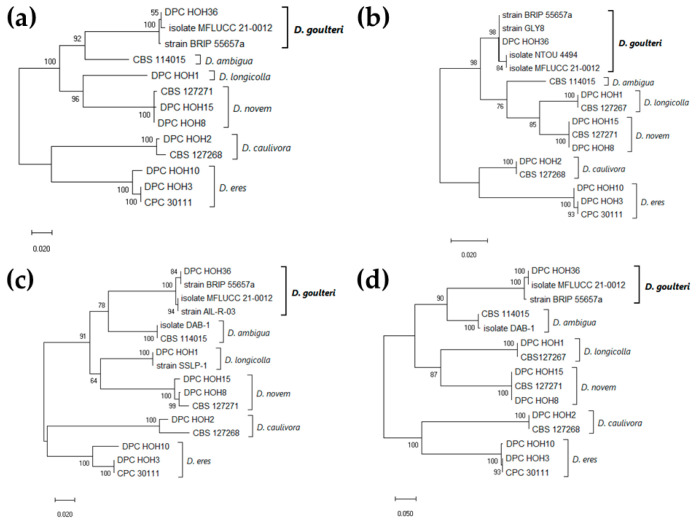
Maximum Likelihood phylogeny identifying isolate DPC_HOH36 as *Diaporthe goulteri*. Trees with the highest log likelihood (lL) are shown. Trees are drawn to scale, with branch lengths indicating the number of substitutions per site. Numbers next to the branches are bootstrapping percentages. (**a**) Combined phylogeny based on 13 concatenated ITS, *TUB*, and *TEF* sequences, 1183 positions in final dataset, lL −4658.69. (**b**) Phylogeny based on 16 ITS sequences, 394 positions in final dataset, lL −1253.80. (**c**) Phylogeny based on 16 *TUB* sequences, 493 positions in final dataset, lL −1581.92. (**d**) Phylogeny based on 15 *TEF* sequences, 324 positions in final dataset, lL −1799.90.

**Figure 2 jof-10-00803-f002:**
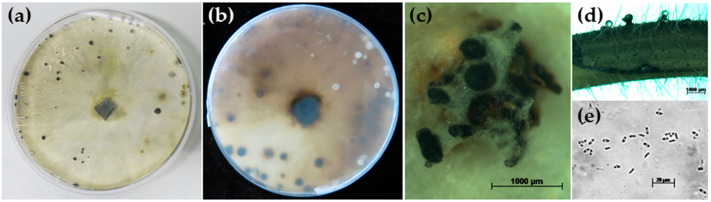
Morphological characteristics of *Diaporthe goulteri* isolate DPC_HOH36. (**a**,**b**) Colony front and back after 40 days on PDA. (**c**) Conidiomata on PDA. (**d**) Pycnidia on soybean stem after three weeks in culture (on water agar). (**e**) α-conidia.

**Figure 3 jof-10-00803-f003:**
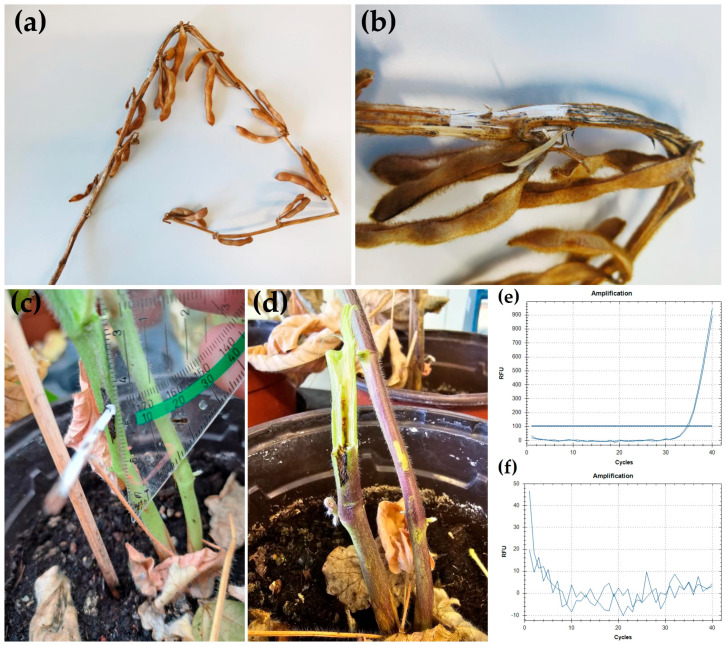
*Diaporthe goulteri* on soybean. (**a**) Soybean plant from which *D. goulteri* isolate DPC_HOH36 was recovered (without rootstock). (**b**) Close-up from (**a**) where the stem is broken open. (**c**) Stem of the soybean plant inoculated with *D. goulteri* isolate DPC_HOH36 using the toothpick method 30 days after inoculation. (**d**) Discoloration of central tissue close to the inoculation site two months after the inoculation. (**e**,**f**) Amplification curves of qPCR assays on DNA sample from plant inoculated with DPC_HOH36 and DNA sample from control plant, respectively.

**Table 1 jof-10-00803-t001:** Strains or isolates of *Diaporthe* spp. and GenBank accession numbers of the ITS (internal transcribed spacer), *TUB* (β-tubulin), and *TEF* (translation elongation factor 1-α) sequences used for the phylogenies in Figure 1.

Species	Designation *	Accession Numbers		
		ITS	*TUB*	*TEF*
*D. ambigua*	CBS 114015	MH862953	KC343978	GQ250299
	Isolate DAB-1		MK463859	MK463861
*D. caulivora*	DPC_HOH2	MK024677	MK161476	MK099094
	CBS 127268	HM347712	KC344013	HM347691
*D. eres*	DPC_HOH3	MK024678	MK161477	MK099095
	DPC_HOH10	MK024685	MK161484	MK099102
	CPC 30111	MG281083	MG281256	MG281604
*D. goulteri*	DPC_HOH36	PQ008930	PQ014385	PQ014381
	Strain BRIP 55657a	KJ197290	KJ197270	KJ197252
	Isolate MFLUCC 21-0012	MW677456	MW680162	MW680164
	Strain AIL-R-03		ON221693	
	Strain GLY8	MF356582		
	Isolate NTOU 4494	MZ422958		
*D. longicolla*	DPC_HOH1	MK024676	MK161475	MK099093
	Strain SSLP-1		HQ333510	
	CBS 127267	HM347700		HM347685
*D. novem*	DPC_HOH8	MK024683	MK161482	MK099100
	DPC_HOH15	MK024690	MK161489	MK099107
	CBS 127271	HM347710	KC344125	HM347695

* Names of strains or isolates for which sequences were deposited in GenBank. Some of these were assigned by the authors who isolated them and deposited the sequences, some represent numbers of culture collections.

## Data Availability

Sequences underlying our analyses were submitted to GenBank, accession numbers are given in the manuscripts. All other data are contained within the manuscript.

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
