# Peer review of "First Report of Diaporthe goulteri on Soybean in Germany"

_jof, 2024, doi:10.3390/jof10110803_

Round 1

Reviewer 1 Report

Comments and Suggestions for Authors

The paper presents the first report of a potential pathogen Diaporthe goulteri on soybean in Germany thus expanding our knowledge on distribution, biology and ecological preferences of the relatively recently detected and described species. Although the research idea and study aim are sound and clear, and the paper is a kind of "short communication" that is easy to read, there are certain shortcomings in the submitted manuscript. In the Materials and Methods section, the sites of sample collection are not adequately described - was it a single soybean field, or several? What are the geographic coordinates of the sampling sites? This is particularly important as the point of detection of D. goulteri must definitely be presented in such a kind of report. And there are many questions regarding sample collection: when the samples have been collected? How many soybean plants have been collected?
The methodology of morphological identification of D. goulteri is not adequately presented - it requires a more thorough description. It is not clear what was the amount and size (dimensions or weight) of plant tissue samples subjected to fungal isolation and direct DNA extraction. How many plants/samples have been screened in total? I think this information is very important. Another important issue - the description of pathogenicity tests also requires revision. When these tests have been carried out? The number of inoculated plants is missing and the sampling procedure from these plants requires better explanation. Have the authors measured the size (length) of the fungus-induced discolorations/necroses on stems? This is very important!

Some parts from the Results section and the legend of Figure 2 may be transferred to the Materials and Methods section - please see my comments in the attached reviewed manuscript file. 

The description of study results is in general well presented, yet some clarifications (especially in presenting the results of pathogenicity tests) would be needed. It is not clear what were tissue reactions in control plants following their inoculation with sterile toothpicks. Moreover, I would strongly suggest adding information about the length of discolorations (necroses?) on the soybean stems following inoculation. Here, it would be good to make sure whether D. goulteri has caused just a discoloration of the invaded soybean tissues or their necrosis. I think this is very important to distinguish in pathogenicity tests! It is not clear how many inoculated plants were subjected for direct DNA extraction (for qPCR).

Legends of Figures 2 & 3 need revision, and some of the images presented in Fig. 3 (c and d in particular) could have been of better quality (resolution) as now they provide little information to the reader.

The first two paragraphs of the Discussion section are in general the repetition of results and should either be removed or rewritten without the repetition of results. The Discussion is rather poor and lacks deeper insights into the ecology/possible pathogenicity of D. goulteri. It is a pity that the performed pathogenicity tests did not allow drawing a clear conclusion whether D. goulteri can cause disease in soybean (i.e., is pathogenic to this species) or not.

In text and figure titles, some abbreviations need explanation. In reference list, some cited papers lack DOI numbers.

More comments, corrections and suggestions can be found in the attached reviewed manuscript file. 

Author Response

The paper presents the first report of a potential pathogen Diaporthe goulteri on soybean in Germany thus expanding our knowledge on distribution, biology and ecological preferences of the relatively recently detected and described species. Although the research idea and study aim are sound and clear, and the paper is a kind of "short communication" that is easy to read, there are certain shortcomings in the submitted manuscript. In the Materials and Methods section, the sites of sample collection are not adequately described - was it a single soybean field, or several? What are the geographic coordinates of the sampling sites? This is particularly important as the point of detection of D. goulteri must definitely be presented in such a kind of report. And there are many questions regarding sample collection: when the samples have been collected? How many soybean plants have been collected?
The methodology of morphological identification of D. goulteri is not adequately presented - it requires a more thorough description. It is not clear what was the amount and size (dimensions or weight) of plant tissue samples subjected to fungal isolation and direct DNA extraction. How many plants/samples have been screened in total? I think this information is very important. Another important issue - the description of pathogenicity tests also requires revision. When these tests have been carried out? The number of inoculated plants is missing and the sampling procedure from these plants requires better explanation. Have the authors measured the size (length) of the fungus-induced discolorations/necroses on stems? This is very important!

Here are a lot of comments and questions that were also raised in the reviewed manuscript pdf file. We chose to answer most of them there.

Some parts from the Results section and the legend of Figure 2 may be transferred to the Materials and Methods section - please see my comments in the attached reviewed manuscript file. 

Responses to these comments can now also be found in the said file.

The description of study results is in general well presented, yet some clarifications (especially in presenting the results of pathogenicity tests) would be needed. It is not clear what were tissue reactions in control plants following their inoculation with sterile toothpicks. Moreover, I would strongly suggest adding information about the length of discolorations (necroses?) on the soybean stems following inoculation. Here, it would be good to make sure whether D. goulteri has caused just a discoloration of the invaded soybean tissues or their necrosis. I think this is very important to distinguish in pathogenicity tests! It is not clear how many inoculated plants were subjected for direct DNA extraction (for qPCR).

See responses in the reviewed manuscript file.

Legends of Figures 2 & 3 need revision, and some of the images presented in Fig. 3 (c and d in particular) could have been of better quality (resolution) as now they provide little information to the reader.

The legends were revised. The resolution of the figures is better in our original files and should be better in the final publication.

The first two paragraphs of the Discussion section are in general the repetition of results and should either be removed or rewritten without the repetition of results. The Discussion is rather poor and lacks deeper insights into the ecology/possible pathogenicity of D. goulteri. It is a pity that the performed pathogenicity tests did not allow drawing a clear conclusion whether D. goulteri can cause disease in soybean (i.e., is pathogenic to this species) or not.

Indeed, we are repeating results here, but we emphasize the comparison to the results by Thompson et al here. We prefer to keep this. In general, there is not much to write here as there are only two publications reporting anything about D. goulteri so far. Testing D. goulteri for pathogenicity to soybean and determining its relevance requires a significant amount of experimentation, including field experiments and studies of incidence and epidemiology. His has not been done so far, which is why this is a short communication. Here we report that D. goulteri occurs on soybean and that it can infect the plant. We cannot and we do not make any other claims.

In text and figure titles, some abbreviations need explanation. In reference list, some cited papers lack DOI numbers.

Now provided.

More comments, corrections and suggestions can be found in the attached reviewed manuscript file. 

Responses to these comments can now also be found in the said file.

Some final remarks:

Your revision is extremely detailed. You must have invested a lot of time into this – helping us to improve our manuscript. We very much appreciate your effort and we followed your recommendations as much as possible. Thank you very much.

Reviewer 2 Report

Comments and Suggestions for Authors

The whole manuscript can be improved by correcting the sentence structure, grammar and more.

Here are few comments:

Line 32-35. I suggest rewriting this sentence.

Line 92: Add and. (One was re-isolation, and the other was….)

Page 98: Please write clearly that you used morphological method to identify the fungus you obtained from the plants (Kochs postulate). Also, I would suggest using your molecular identification method as well.

Line 159: make the sentence clear.

 Line 158-164: Rewrite this whole paragraph.

Line 160-161: What are you trying to explain? Endophyte vs Ectophyte? Most plant pathogens should be endophyte so that affect the plants inter or intra cellularly. Please clarify. You also mentioned “discoloration of central tissue” in fig 2, so it should be endophytes.

Line 132: “which parts of the plant are invaded”… What other plant parts did you try to infect with this pathogen. It is not mentioned in MM section.  

Comments on the Quality of English Language

English language can be improved. Some of the sentences are ambiguous and not very clearly written.

Author Response

The whole manuscript can be improved by correcting the sentence structure, grammar and more.

Thank you for your suggestions. Many parts of the manuscript were rewritten, based on your suggestions and also based on the suggestions of the other reviewers.

Here are few comments:

Line 32-35. I suggest rewriting this sentence.

The sentence has been changed.

Line 92: Add and. (One was re-isolation, and the other was….)

Was changed.

Page 98: Please write clearly that you used morphological method to identify the fungus you obtained from the plants (Kochs postulate). Also, I would suggest using your molecular identification method as well.

We think this is clear. We now also mention Koch’s postulate. We did identify D. goulteri in the plant using qPCR, so using qPCR on the fresh isolate did not seem necessary. This has been done since but we do not think that it is necessary to write here.

Line 159: make the sentence clear.

 Line 158-164: Rewrite this whole paragraph.

Line 160-161: What are you trying to explain? Endophyte vs Ectophyte? Most plant pathogens should be endophyte so that affect the plants inter or intra cellularly. Please clarify. You also mentioned “discoloration of central tissue” in fig 2, so it should be endophytes.

Yes. But more importantly we argue that the plant that we found in the field was truly infected with D. goulteri as opposed to the possibility that we mistakenly make the association between D. goulteri and soybean. We shortened this and moved some of the arguments to the last part of the discussion.

Line 132: “which parts of the plant are invaded”… What other plant parts did you try to infect with this pathogen. It is not mentioned in MM section. 

Line 162? We changed the wording here. What we wanted to know is whether the fungus would spread through the whole plant from our point of inoculation or whether it would be restricted to a smaller area. Since we found the fungus in a seed it might have been expected that seeds would also be infected in the inoculated plant. This did not happen in this experiment though we found further growth of the fungus in later experiments.

Reviewer 3 Report

Comments and Suggestions for Authors

This manuscript describes the identification of Diaporthe goulteri on soybean in southern Germany. The authors collected a Diaporthe spp (DPC_HOH36) from a soybean plant and identified it as D. goulteri based on molecular evidence (sequences of ITS, TUB, and TEF genes) and morphological characteristics. Furthermore, they showed that the fungus was able to infect soybean plants by artificial inoculation with mycelum. As this was the first report that D. goulteri was able to infect soybean, the story should be of interest to community of plant pathologists and soybean growers. Although it is of significance, the manuscript still needs to be improved in several aspects.

Major

1, It is not clear whether not the symptoms shown on soybean plants artificially inoculated were compatible to those observed in the field (Figure 3, line 168). As spread of the disease is most likely through conidial spores, I would suggest that inoculation of soybean plants with conidia also be performed.

2, What was the proportion of D. goulteri among Diaporthe species in the soybean field in southern Germany?

3, The manuscript was not well organized and written. Descriptions of symptoms (Figure 3 a & b, line 168) should be placed first in the Result section. Re-isolation of the fungus from the artificially inoculated soybean plants should be clearly stated.

Minors

1, Figure 2d: image of pycnidia is not obvious and clear; Figure 2e: image of α-conidia was not easily seen. Use a larger magnification.

2, Figure 3 e & f, line 168: Amplification curves of qPCR is not much informative. A regular PCR would be better. I suggest replacing Figure 3 e & f with a PCR gel image.

3, Text contains many errors or inappropriate expresions, e.g.,

Lines 29-31: …and studies concerned with this regularly discover new species belonging to the genus. ... This group [12] isolated the species from a sunflower (Helianthus annuus) seed, described it and named it.

Line 49: 2.2.1. From cultures using morphology, sequencing and phylogenetic analysis

Line 50, 51: 20 days-old, …, 7 days-old

Line 52, 54: published by [14], described by [7].

Line 54: BLASTed, is this a correct word?

Line 64: 2.2.2. “Directly from infected plant parts using qPCR” is not an appropriate heading.

Lines 84-85: For this, autoclaved toothpicks were ...

Lines 99-100: The developing fungal colonies were determined as …

Line 108: the qPCR assay developed by [22],

Line 115: sequences annotated with either D. goulteri or D. ambigua

Lines 152-153: “The α-conidia were fusiform to cylinder-shaped, biguttulate, hyaline, and 5.19 to 8.06 x 1.92 to 2.88 μm in diameter”, delete “in diameter”.

Line 165: Originating from where the toothpicks were inserted into the soybean stem, we could observe dark discoloration. …

Line 186, “[12] described D. goulteri colonies growing on PDA and OMA with relatively…”,

Line 195: “[12] isolated D. goulteri from a seed of Helianthus annuus”

Comments on the Quality of English Language

Some sentences are not easy to follow. Extensive editing is required.

Author Response

This manuscript describes the identification of Diaporthe goulteri on soybean in southern Germany. The authors collected a Diaporthe spp (DPC_HOH36) from a soybean plant and identified it as D. goulteri based on molecular evidence (sequences of ITS, TUB, and TEF genes) and morphological characteristics. Furthermore, they showed that the fungus was able to infect soybean plants by artificial inoculation with mycelum. As this was the first report that D. goulteri was able to infect soybean, the story should be of interest to community of plant pathologists and soybean growers. Although it is of significance, the manuscript still needs to be improved in several aspects.

Thank you for reviewing and thank you for your comments. We made several changes to the manuscript. Because of time restrictions we could deal with most of your minor comments but could do little about the major ones. Nevertheless, we hope that we could address your concerns.

Major

1, It is not clear whether not the symptoms shown on soybean plants artificially inoculated were compatible to those observed in the field (Figure 3, line 168). As spread of the disease is most likely through conidial spores, I would suggest that inoculation of soybean plants with conidia also be performed.

As you correctly observed, the inoculation method that we chose does not optimally reflect natural infection. We chose this method because it is quick and easy to perform. It is also true that the symptoms we observed on the inoculated plant were different from those of the plant that D. goulteri was isolated from; the biggest difference lies in the fact that the seeds, from which D. goulteri was isolated, remained uninfected in the inoculated plant. This is definitely due to the inoculation method. As we are writing at the end of the paper, more experiments for testing the pathogenicity will be necessary. This is part of the reason why we chose Short Communication for publishing our results; for this, our evidence showing the association of D. goulteri to soybean should be sufficient in our opinion.

2, What was the proportion of D. goulteri among Diaporthe species in the soybean field in southern Germany?

A very good question that we would like to answer in the near future. For this purpose, we established a species specific qPCR detection method for D. goulteri, as described in this manuscript. But testing of sufficient samples to determine the incidence of D. goulteri still has to be done, so these data cannot be part of this paper.

3, The manuscript was not well organized and written. Descriptions of symptoms (Figure 3 a & b, line 168) should be placed first in the Result section. Re-isolation of the fungus from the artificially inoculated soybean plants should be clearly stated.

We did consider making Figure 3 into two figures. But in the end we decided that keeping all data considering Koch’s postulates together in one figure. Also we think that a Short Communication should not have too many figures.

We added to the description of the re-isolation in the Methods section.

Minors

1, Figure 2d: image of pycnidia is not obvious and clear; Figure 2e: image of α-conidia was not easily seen. Use a larger magnification.

Using this magnification gave the best pictures. We noticed that all figures in the PDF manuscript have less resolution than our original figures. In our original figure the image of the α-conidia is quite reasonable. We expect that this will no longer be an issue once the figures are inserted in full resolution by the layout staff in MDPI.

2, Figure 3 e & f, line 168: Amplification curves of qPCR is not much informative. A regular PCR would be better. I suggest replacing Figure 3 e & f with a PCR gel image.

qPCR is a method frequently used for diagnosis of pathogens. For quantification purposes it is qPCR that is preferred. In this case, where only a yes or no answer was needed the qPCR amplification curve does not provide any additional information as compared to a gel image but the gel image does not provide any information that cannot be found in the amplification curve, either.

3, Text contains many errors or inappropriate expresions, e.g.,

Lines 29-31: …and studies concerned with this regularly discover new species belonging to the genus. ... This group [12] isolated the species from a sunflower (Helianthus annuus) seed, described it and named it.

We rephrased this sentence.

Line 49: 2.2.1. From cultures using morphology, sequencing and phylogenetic analysis

We rephrased this heading.

Line 50, 51: 20 days-old, …, 7 days-old

Removed the dashes, we considered this writing to be better understandable.

Line 52, 54: published by [14], described by [7].

Yes, this does not read very nice, but writing out the names et al. goes against the idea of the numbered citation style.

Line 54: BLASTed, is this a correct word?

Rephrased.

Line 64: 2.2.2. “Directly from infected plant parts using qPCR” is not an appropriate heading.

Rephrased

Lines 84-85: For this, autoclaved toothpicks were ...

Rephrased

Lines 99-100: The developing fungal colonies were determined as …

Rephrased

Line 108: the qPCR assay developed by [22],

Yes, this does not read very nice, but writing out the names et al. goes against the idea of the numbered citation style.

Line 115: sequences annotated with either D. goulteri or D. ambigua

Lines 152-153: “The α-conidia were fusiform to cylinder-shaped, biguttulate, hyaline, and 5.19 to 8.06 x 1.92 to 2.88 μm in diameter”, delete “in diameter”.

Changed.

Line 165: Originating from where the toothpicks were inserted into the soybean stem, we could observe dark discoloration. …

Rephrased.

Line 186, “[12] described D. goulteri colonies growing on PDA and OMA with relatively…”,

Rephrased.

Line 195: “[12] isolated D. goulteri from a seed of Helianthus annuus”

Rephrased.

Reviewer 4 Report

Comments and Suggestions for Authors

This report is clear and quite believable.  I believe it is publishable as is.

Author Response

This report is clear and quite believable.  I believe it is publishable as is.

Thank you very much for the recommendation.

Round 2

Reviewer 3 Report

Comments and Suggestions for Authors

The revision has addressed many of my concerns. I believe the manuscript could be further improved if the following issues could be resolved or clarified.

The authors collected soybean samples with Diaporthe infection symptoms and screened them by qPCR assay for presence of D. caulivora, D. eres, D. longicolla, or D. novem. They found some of the samples were negative in qPCR assay but Diaporthe spp could be isolated. One of these Diaporthe spp, DPC_HOH36, was further identified to be D. goulteri. To smooth the evidence chain, more detailed information should be given.

1, Field samples were tested for the presence of D. caulivora, D. eres, D. longicolla, or D. novem (Lines 84-85). What was the percentage of samples positive for these species among the total samples tested? Among the negative samples, how many were successful in isolation of Diaporthe spp? From these Diaporthe spp, how many were identified as D. goulteri? Was DPC_HOH36 the only isolate gained?

2, Lines 87-96: detection of D. goulteri was performed. Please specify the source of sample- field sample or culture mycelium?

Author Response

The revision has addressed many of my concerns. I believe the manuscript could be further improved if the following issues could be resolved or clarified.

Thank you once more for reviewing!

The authors collected soybean samples with Diaporthe infection symptoms and screened them by qPCR assay for presence of D. caulivora, D. eres, D. longicolla, or D. novem. They found some of the samples were negative in qPCR assay but Diaporthe spp could be isolated. One of these Diaporthe spp, DPC_HOH36, was further identified to be D. goulteri. To smooth the evidence chain, more detailed information should be given.

1, Field samples were tested for the presence of D. caulivora, D. eres, D. longicolla, or D. novem (Lines 84-85). What was the percentage of samples positive for these species among the total samples tested? Among the negative samples, how many were successful in isolation of Diaporthe spp? From these Diaporthe spp, how many were identified as D. goulteri? Was DPC_HOH36 the only isolate gained?

As mentioned in Results (line 124), our sampling is supposed to be part of a survey to determine the incidence of the different Diaporthe spp. in Germany. We are planning to publish the results of this survey in detail in another paper. The survey is still ongoing and the results are not yet evaluated.

Following your recommendation, we added an estimate from a preliminary evaluation in Results following line 127. As we sampled symptomatic plants, the percentage of positives was relatively high.

As of now, we isolated D. goulteri once. We do not think this needs to be stressed again. We already state that one isolate was identified as D. goulteri.

Besides DPC_HOH36 we did gain other isolates. Some of them were actually from samples that tested positive for the mentioned species and we performed them to gain fresh isolates for these species. We also gained isolates of other species. We do not want to include this information here because we think this detracts from our central message and because we want to publish this in another publication.

2, Lines 87-96: detection of D. goulteri was performed. Please specify the source of sample- field sample or culture mycelium?

As this is Methods we did not think it was necessary to provide this information here. In principle, the assay can be used to identify the species in culture, in plant samples, and in soil samples. Here we used it to detect D. goulteri in the inoculated plants of our pathogenicity assay. This is mentioned in results, line 200ff.

We added “used here […] in plant samples” in Methods line 87.